# Comparative analysis of syngeneic mouse models of high-grade serous ovarian cancer

David P. Cook [1,2 ✉], Kristianne J. C. Galpin [1,2], Galaxia M. Rodriguez[1,2], Noor Shakfa [3],
Juliette Wilson-Sanchez[3], Maryam Echaibi[1,2], Madison Pereira [4], Kathy Matuszewska[4], Jacob Haagsma[5],
Humaira Murshed[1,2], Alison O. Cudmore[1,2], Elizabeth MacDonald[1], Alicia Tone[6], Trevor G. Shepherd[5],
James J. Petrik[4], Madhuri Koti[3] & Barbara C. Vanderhyden [1,2]

Ovarian cancers exhibit high rates of recurrence and poor treatment response. Preclinical models that recapitulate human disease are critical to develop new therapeutic approaches. Syngeneic mouse models allow for the generation of tumours comprising the full repertoire of non-malignant cell types but have expanded in number, varying in the cell type of origin, method for transformation, and ultimately, the properties of the tumours they produce. Here we have performed a comparative analysis of high-grade serous ovarian cancer models based on transcriptomic profiling of 22 cell line models, and intrabursal and intraperitoneal tumours from 12. Among cell lines, we identify distinct signalling activity, such as elevated inflammatory signalling in STOSE and OVE16 models, and MAPK/ERK signalling in ID8 and OVE4 models; metabolic differences, such as reduced glycolysis-associated expression in several engineered ID8 subclones; and relevant functional properties, including differences in EMT activation, PD-L1 and MHC class I expression, and predicted chemosensitivity. Among tumour samples, we observe increased variability and stromal content among intrabursal tumours. Finally, we predict differences in the microenvironment of ID8 models engineered with clinically relevant mutations. We anticipate that this work will serve as a valuable resource, providing new insight to help select models for specific experimental objectives.

[1] Cancer Therapeutics Program, Ottawa Hospital Research Institute, Ottawa, ON, Canada. [2] Department of Cellular and Molecular Medicine, University of Ottawa, Ottawa, ON, Canada. [3] Queen's Cancer Research Institute, Kingston, ON, Canada. [4] Department of Biomedical Sciences, Ontario Veterinary College, University of Guelph, Guelph, ON, Canada. [5] The Mary & John Knight Translational Ovarian Cancer Research Unit, Lawson Health Research Institute, London, ON, Canada. [6] Ovarian Cancer Canada, 145 Front St E #205, Toronto, ON, Canada. ✉email: david.cook@uottawa.ca

Approximately 1% of North Americans born with ovaries will lose their lives to ovarian cancer. Despite decades of research, the standard treatment for many of these individuals remains unchanged. The discovery of synthetic lethality upon exposure of homologous recombination-deficient (HRD) malignant cells to PARP inhibition provided a targeted therapy for approximately 50% of patients with high-grade serous ovarian cancer (HGSOC)—the most common and most lethal form of ovarian cancer[1–3]. However, many patients with HRD HGSOC fail to respond to this treatment or develop resistance following prolonged treatment[4]. Developing targeted therapies for the remaining 50% of patients has been challenging due to both the phenotypic and genetic diversity of malignant populations[5,6]. This has put an increased demand on preclinical models to faithfully recapitulate this complexity.

Preclinical models for HGSOC are fortunately abundant. Human-derived models sample genetic diversity of the disease[7,8] but are incapable of being used to model tumours comprising the complete repertoire of non-malignant stromal cells. Successful culture of tumour explants can maintain malignant-stromal interactions ex vivo, but these models are not amenable to long-term propagation[9,10]. Establishing patient-derived xenografts in humanised mouse models with adoptive transfer of autologous leucocytes restores components of stromal interactions[11], but it is unclear how accurately these models mirror the progression and therapy response of the native tumour. Thus, human-derived models may be ideal for cell autonomous properties of the malignant population but are limited in their ability to model cancer progression and the complex interactions within the tumour microenvironment (TME).

Syngeneic mouse models are powerful resources that allow for the generation of tumours in immunocompetent mice. Their ability to model the genetics of human disease is limited, but gene editing strategies can be used to engineer clinically relevant mutations, such as the nearly ubiquitous TP53 mutation observed in 96% of HGSOC tumours[1]. However, the complex genomic rearrangements observed in HGSOC have been more challenging to model. Despite this limitation, orthotopic tumours can be generated from these models with histological features similar to human disease, making them a prime resource for studying both the evolution of a complex TME within the ovary and the testing of therapeutics that depend on interactions within it.

Various syngeneic models of HGSOC have been developed, but the strategies to establish them—the cell type of origin, the strategy for malignant transformation, and the engineering of relevant mutations—have varied. Differences in the experimental progression of these models have been reported, including differences in growth rate, TME composition, and sensitivity to treatment[12–16]. Thus, there is a need to comprehensively compare these models in order to understand their inherent differences. In this study, we have performed transcriptomic profiling of 22 syngeneic models, as well as tumours derived from select models. We evaluate inherent differences associated with cell type of origin, the phenotypic divergence associated with spontaneous transformation from prolonged in vitro culture, and the impact of engineering clinically relevant mutations. We explore how these phenotypes give rise to tumours with unique TMEs, such as intrabursal STOSE tumours with low stromal content or the lymphocyte-rich ID8-Trp53$^{−/−}$ model. Together, this work provides insight into the properties of diverse syngeneic models that can inform the selection of appropriate models for subsequent research and therapeutic testing.

## Results

**Transcriptomic profiling of syngeneic mouse models of high-grade serous ovarian cancer.** To develop a resource of transcriptomic data that could provide further insight into models of HGSOC, we performed RNA-seq on a collection of mouse ovarian cancer cell lines (Fig. 1a). This collection comprises models from both oviductal (OVE/MOE[17]) and ovarian surface epithelium (OSE), spontaneously transformed (STOSE[18] and ID8[19]) models, secondary lines derived from ascites, and derivative lines engineered with clinically relevant mutations in tumour suppressor genes or constitutive activation of oncogenes[20–22]. For select models, we also sequenced RNA from tumours derived from either intrabursal (IB) or intraperitoneal (IP) injection of the cells, allowing us to evaluate how properties of the models and TME may affect features of the resultant tumours (Fig. 1a). All evaluated models were initially derived from primary cultures of non-tumourigenic ovarian or oviductal epithelium. Tumorigenicity was acquired spontaneously through long-term propagation (STOSE and ID8) or repression of the tumour suppressors Trp53 (OVE4 and OVE16 derivatives) or Pten (MOE).

Across all models, we found transcriptional profiles of biological replicates ($n = 3–5$/model) were strongly correlated (Fig. 1b). Unsurprisingly, bulk profiles from tumour samples shared little similarity to those of pure cell lines (Fig. 1c). They did, interestingly, cluster according to whether they were generated through IP or IB injection of cells, likely reflecting inherent differences in TMEs of these models. In general, IB tumours were more variable, even among replicates (Fig. 1b). To evaluate differences in the TME between these two tumour sites, we used CIBERSORTx[23] to decompose the bulk RNA-seq profiles and predict the relative proportions of malignant and non-malignant cell types. The inferred contribution of non-malignant stroma to IB tumours was higher than in IP tumours in all models except STOSE IB tumours, which had notably high malignant purity (Fig. 1d). As such, differences that emerge in each tumour's microenvironment during its development will have a greater impact on the bulk profiles, contributing to the increased variability among IB samples.

**Cell line models exhibit distinct signalling, metabolic, and functional properties.** Principal component analysis (PCA) of the cell line samples highlighted the impact of the models' cell of origin, with the first principal component (PC) separating OSE- and OVE-derived models (Fig. 2a). Despite factors such as long-term in vitro culture and genetic manipulation, an imprint of the cells' original identity persisted in culture: various genes uniquely associated with OSE and OVE identity were among the top loaded genes for PC1, including Pax8 and Krt7 in OVE lines and Krt19 and Amhr2 in OSE-derived lines (Fig. 2b). This distinction based on cell of-origin has also been previously observed in both engineered organoid models from OSE and OVE tissue[24] and genetically engineered mouse models of HGSOC[25], suggesting that although models from both tissues can form tumours with HGSOC characteristics, phenotypic differences of the malignant compartment exist. We performed gene set enrichment analysis (GSEA) on PC1 loading-ranked genes to identify such properties. Consistent with their cell of origin, OVE lines appear more differentiated, with elevated expression of epithelial signatures and MAPK/ERK signalling (Fig. 2c)[26]. OSE-derived lines, however, were associated with a more mesenchymal phenotype (Fig. 2c).

To further explore biological properties differing among the various models, we collected transcriptional signatures associated with multiple signalling pathways, biological function, and metabolic processes (Supplementary Data 2)[27–32]. We calculated rank-based signature scores for each gene set and compared the relative activity between models (Fig. 2d). This revealed features that are particularly relevant for the selection of appropriate models for preclinical studies. For example, the ID8 models had

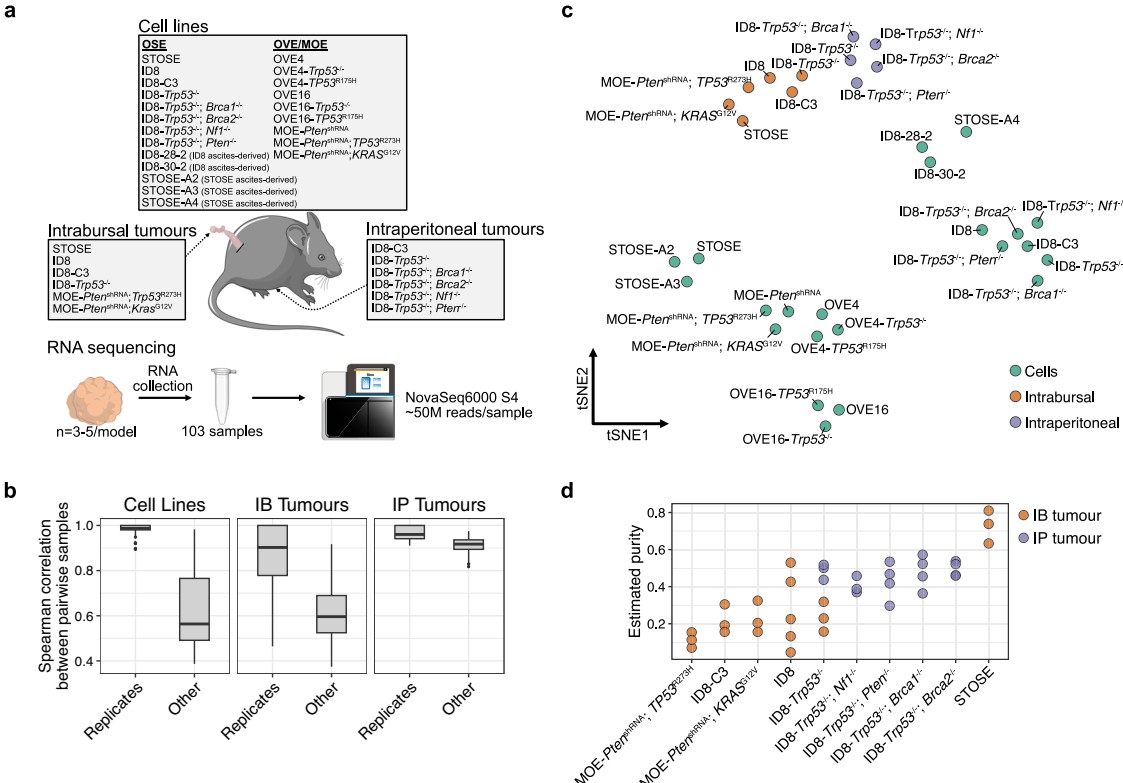

**Fig. 1 Transcriptional profiling of diverse syngeneic mouse models of HGSOC. a** Schematic of the study design. RNA-seq was performed on all listed models and 3–5 replicates were included for each model. **b** Comparison of Spearman correlation coefficients of transcriptional profiles both within and between models for cell lines, IB tumours, and IP tumours. Expression profiles of biological replicates ($n = 3$–5 per model) were averaged prior to dimensionality reduction. **c** tSNE embedding of the replicated-averaged transcriptional profile of each model. **d** Tumour purity estimates from IB and IP tumours based on the estimated proportion of malignant cells predicted by deconvolving bulk RNA-seq profiles with CIBERSORTx. Cell type profiles used from deconvolution were derived from scRNA-seq data of STOSE and ID8 tumours[13].

higher proliferative activity (E2F targets), with elevated oxidative phosphorylation activity and MTORC1 signalling, but they lacked activity of various cytokine-associated pathways (TGFb, TNFa, NFkB). These features may contribute to a predicted increase in sensitivity to chemotherapy, but these cells also had reduced expression of MHC class I components and expressed higher levels of the immunosuppressive factor PD-L1, which may predict poorer outcomes and/or decreased sensitivity to immunotherapy. In contrast, JAK-STAT, TGFb, and TNF signalling are elevated in OVE16 and STOSE cell lines, which may be ideal to model tumours with high levels of inflammation and chemoresistance (Fig. 2d). While these inferences are strictly based on expression levels of relevant genes, the predicted decrease in chemosensitivity of the MOE (originally derived from OVE4) cell lines relative to ID8 lines is consistent with survival data from carboplatin-treated IP tumours (Fig. 2e). These findings are also in line with previous work demonstrating that the ID8-*Trp53*$^{-/-}$; *Brca2*$^{-/-}$ model is more sensitive to chemotherapy than ID8-*Trp53*$^{-/-}$ tumours[21].

**TP53^{R175H} partially restores TP53 function and activates several novel pathways**. We next examined the impact of nonsense or the common R175H missense mutations on the transcriptional profiles of several models. All *Trp53*-null models were generated by targeting the endogenous locus with CRISPR-Cas9. *Trp53*-null OVE4 and OVE16 cells were additionally transduced with lenti-virus carrying the human *TP53*$^{R175H}$. We first applied PROGENy to infer the activity of p53 signalling based on the expression of perturbation-responsive genes[27]. As expected, relative p53 activity was reduced in all models with null mutations (Fig. 3a).

Transcriptional changes associated with null mutations, however, were quite variable among models, with many unique effects in each line (Fig. 3b). For example, only 25% of downregulated genes were shared between the two OVE models, and genes upregulated in ID8 cells had a larger overlap with those down-regulated in OVE cells than upregulated (Fig. 3b). Several genes were implicated in all three models with null mutations, however. Only *Map1a, Rcn3,* and *Prr5l* were upregulated in all three models, whereas 24 genes were downregulated in all three, including *Ngf, Cdkn1a, Areg, Tspan7, Itgb6,* and *Cxcl17* (Fig. 3c, d).

Hoadley et al.[33] previously examined p53 signalling activity in human tumours harbouring *TP53* missense mutations and, using the PARADIGM method to infer signalling activity, found that the majority of missense mutations in HGSOC result in decreased activity relative to several wild-type tumours. In contrast, we observed that expression of *TP53*$^{R175H}$ in *Trp53*-null OVE4/16 cells restored the inferred p53 activity (Fig. 3a). As this may reflect differences in the gene signatures and methods for activity inference, we next evaluated the extent to which *TP53*$^{R175H}$ restored effects caused by disrupting the endogenous *Trp53* locus. In both OVE4 and OVE16, *TP53*$^{R175H}$ recovered the expression of a subset of the genes repressed following *Trp53* deletion (Fig. 3e, f). Just as *Trp53* deletion has unique effects on the various lines, the specific genes recovered by *TP53*$^{R175H}$ varied between lines. OVE4 cells reactivated the expression of various genes involved in damage responses (Fig. 3e), which may relate to the previously observed relationship between the R175H mutation and apoptosis signalling in these cells[34]. While the expression of mutant p53 had minimal novel effects (ie. expression changes

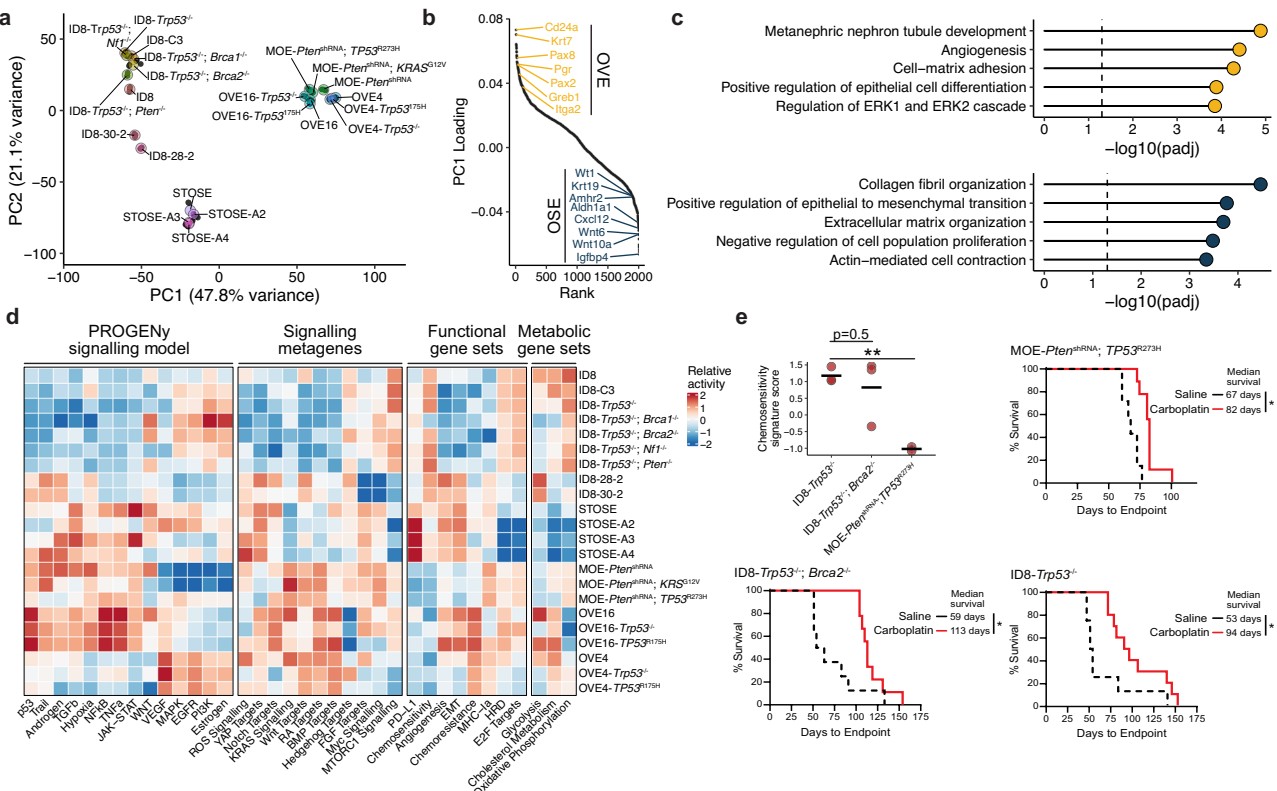

**Fig. 2 Comparison of cell line models for HGSOC. a** PCA embedding of the transcriptional profiles of cell line models. Black points represent individual ($n = 3$–5 per model) and coloured points reflect the average coordinates of model replicates. **b** Ranked gene loadings for the first principal component, which separates OSE and OVE-derived cell lines. Select highly ranked genes are shown. **c** Over-enrichment analysis of GO terms among the top and bottom 200 ranked genes. *P*-values were adjusted using the Benjamini–Hochberg method. **d** Relative (*Z*-score transformed) gene set scores among cell line models. Relative scores were computed for each replicate prior to averaging scores for each model. The source and genes associated with each gene set are listed in Supplementary Data 2. **e** Relative expression of a chemosensitivity signature[31] from RNA-seq samples (top left; $n = 3$ per model; ** $p < 0.01$). Survival Kaplan–Meier plots of FVB/N (MOE-*Pten*shRNA; *TP53*R273H) and C57BL/6 tumour-bearing mice (ID8-*Trp53*−/−;*Brca2*−/−, ID8-*Trp53*−/−), treated with 20 mg/kg carboplatin (red line) or saline (black dotted line). Curves represent $n = 10$ mice per group per model. Log-rank (Mantel-Cox) *$p < 0.05$.

of genes unaffected by *Trp53* deletion in parental cells) in OVE4 cells, mutant expression activated various metabolic and immune-related genes that were unaffected by deletion of wild-type *Trp53* (Fig. 3e). In both models, *TP53*R175H failed to recover the expression of several hundred genes repressed following the deletion of the endogenous *Trp53* (Fig. 3e, f). While many of the specific genes varied, failure to re-activate several features of epithelial differentiation was common, such as the maintained suppression of various cell adhesion genes (e.g., *Cldn1*, *Cldn4*, *Cdh1*) or genes associated with EGF signalling (Fig. 3e, f). This putative dedifferentiation may relate to the various phenotypic properties of cells with *TP53*R175H mutations, including enhanced metastatic capacity, stemness, and drug resistance[35].

**Spontaneous transformation leads to highly divergent phenotypes in STOSE and ID8 models.** The STOSE and ID8 models of HGSOC were both derived from prolonged culture of OSE cells that led to progressive aneuploidy and, ultimately, the capacity to form malignant tumours[18,19] (Fig. 4a). It has been demonstrated that these models produce tumours with distinct microenvironments[13], but beyond each model having a unique constellation of genomic aberrations, the phenotypic differences of these lines have yet to be characterised. To begin to address this, we evaluated genes differentially expressed in the STOSE and ID8 cell lines. Despite being derived from the same cell type—albeit different mouse strain—we observed massive divergence in

their phenotypes, with 5115 differentially expressed genes ($p < 0.05$; |logFC| > 1) (Fig. 4b). We evaluated GO terms enriched in the genes associated with each line and found that STOSE were associated with mesenchymal [extracellular matrix (ECM) organisation, migration] and immunoregulatory terms, whereas ID8 cells were primarily characterised by higher expression of various metabolic pathway components (Fig. 4c, Supplementary Fig. 1a).

Given the challenges of interpreting gene set over-representation with such disparate expression profiles, we evaluated several specific properties of these cells. First, we inferred signalling and transcription factor activity in the two models. Consistent with the mesenchymal and immunoregulatory GO term enrichment, conserved targets of many signalling pathways were preferentially activated in STOSE, including TGFb, TNFa, NFkB, and MAPK signalling (Fig. 4d). Supporting the enriched GO terms and pathway activity inference, we found that STOSE cells also had increased levels of phosphorylated ERK, reflecting the higher activity of MAPK signalling (Supplementary Fig. 1b). In a scratch wound assay, STOSE cells are also more migratory than ID8 (Supplementary Fig. 1c). Related transcription factors (e.g., Smad3/4, Stat3, Twist1, Zeb1) were also predicted to have higher activity in STOSE (Fig. 4e). Evidence of signalling patterns specific to ID8 cells was limited. Wnt targets were more highly expressed, along with transcriptional activity of the epithelium-associated transcription factors Foxa1 and Foxo1,

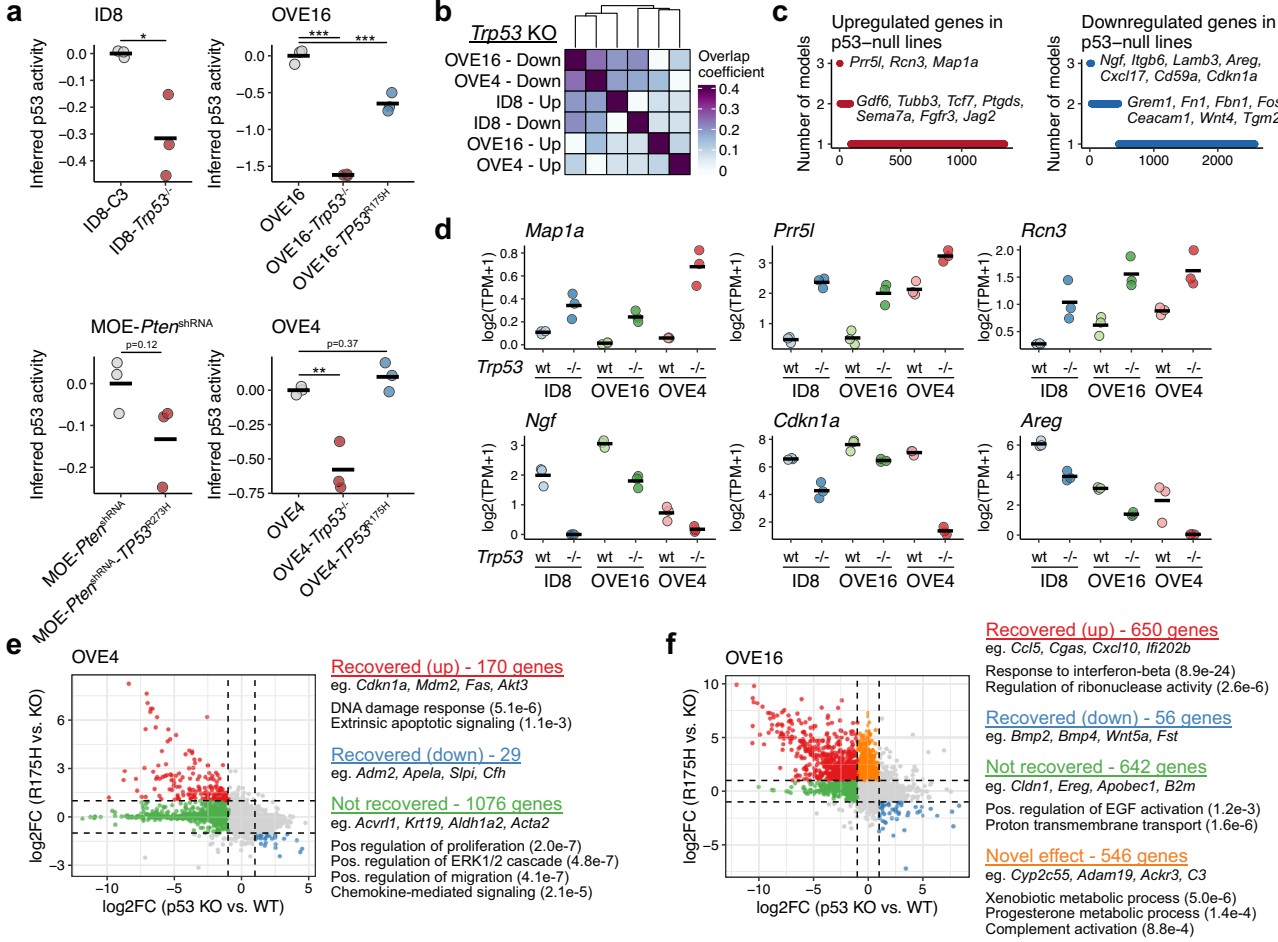

**Fig. 3 Common and divergent effects of null and missense p53 mutations. a** Inferred p53 signalling activity in lines with wild type, nonsense, and missense p53. Activity values are Z-scores derived by PROGENy[27]. **b** Pairwise intersection of differentially expressed genes ($p < 0.05$, |log2FC| > 1; $n = 3$) associated with $Trp53^{-/-}$ mutations relative to parentals lines. **c** Frequency of genes upregulated or downregulated upon $Trp53$ deletion compared to parental lines. **d** Expression values of select genes commonly upregulated (*Map1a, Prr5l, Rcn3*) or downregulated (*Ngf, Cdkn1a, Areg*) following $Trp53$ deletion. Expression values reflect log-transformed transcripts per million (logTPM). **e** Comparison of expression patterns in OVE4 cells following $Trp53$ deletion (p53 KO vs. wild-type) or re-expression of the missense $TP53^{R175H}$ (R175H vs. KO). Select genes and GO terms ($p$-values included in parentheses) are included. **f** Same as (**e**) for OVE16 cells.

and Srebf2, which controls cholesterol homoeostasis, in line with metabolic changes we observed more globally (Fig. 4e).

The observation of mesenchymal properties in STOSE cells led us to evaluate the expression of ECM components in the two models (Fig. 4f). Many components were reliably detected but were not differentially expressed between the two models. However, STOSE cells had elevated expression of a variety of collagens (*Col1a1, Col5a3, Col4a4, Col3a1*, and others) and ECM glycoproteins, including the canonical mesenchymal marker *Fn1*. Notably fewer ECM components were preferentially expressed in ID8 cells (e.g., *Col11a1, Col17a1, Igfbp7, Fbln2*) (Fig. 4f).

To further explore differences in the expression of immunoregulatory factors, we evaluated the expression of cytokines, chemokines, and their receptors in the two models (Fig. 4g). STOSE cells expressed higher levels of diverse chemokines and cytokines, including *Il6, Il18, Tgfb1, Ccl7*, and *Csf1*. Similar to ECM components, fewer factors were more dominantly expressed in ID8 cells. They did, however, express higher levels of *Il15, Cxcl16*, and various cytokine/chemokine receptors (*Ccr7, Ccr4*, and *Il3ra*) (Fig. 4g). We predicted that the imbalanced expression of immunoregulatory factors may lead to differences in the immune infiltration. Using the estimated cell-type proportions from the deconvolution of bulk RNA-seq from the tumours of

these cells, we found that intrabursal injection of STOSE cells yielded higher purity tumours with a reduced proportion of immune cell infiltration (Fig. 1d). Among the non-epithelial fraction in both models, macrophages were more prevalent in STOSE (Fig. 4h), consistent with their elevated chemokine expression. These inferences about the TME of STOSE tumours and their more immunoregulatory phenotype compared to ID8 cells also match the recent flow cytometry-based immunophenotyping and cytokine profiling of these models[13].

**Secondary cell lines derived from ascites stably activate mesenchymal programs.** Collecting metastatic cells from mouse tumour models for the purpose of deriving aggressive sub-lines is a common strategy. Tumours from these sub-lines often progress more rapidly and are more metastatic than parental models. In our cohort of samples, we have included ascites-derived lines from both STOSE and ID8 intrabursal tumours to evaluate how they deviate from their parental lines and to determine if independent lines derived from ascites acquire common traits (Fig. 5a).

We used PCA to evaluate the relationship between ascites lines and their parental cell lines. For both STOSE and ID8 lines, the

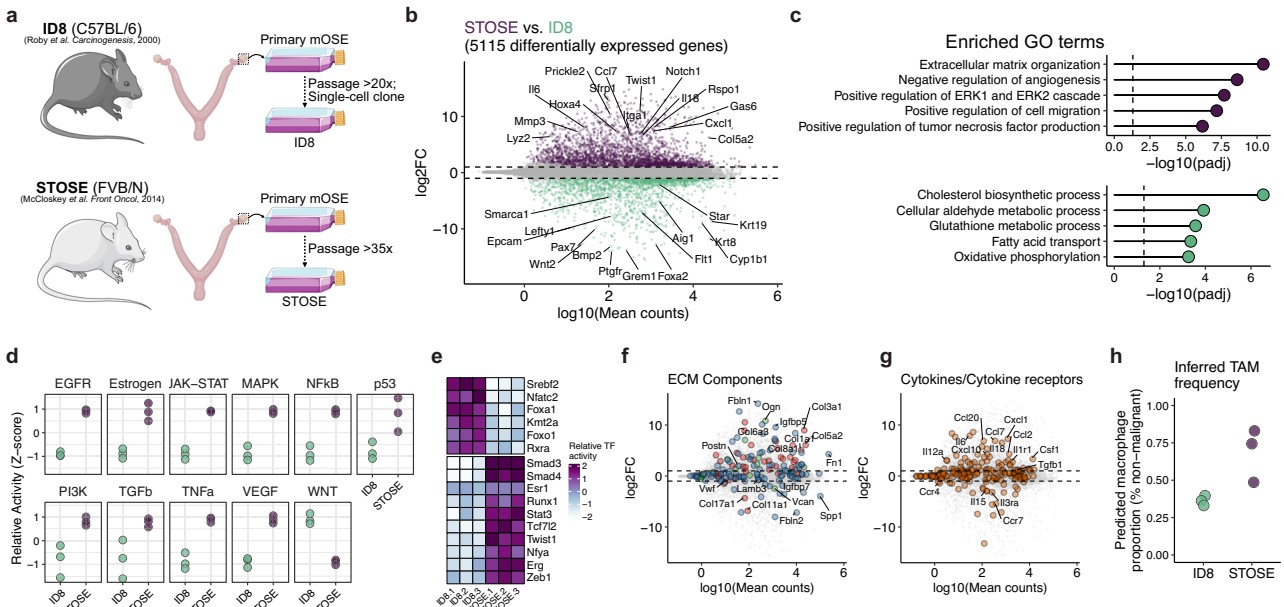

**Fig. 4 Divergence of spontaneously transformed ovarian surface epithelium. a** Schematic demonstrating the establishment of two spontaneously transformed models of HGSOC (STOSE and ID8). **b** Differentially expressed genes between STOSE and ID8 cells. The x-axis reflects the average quantification (mean counts) for each gene and the y-axis shows the log2 fold change (log2FC) between the two models. The dashed line reflects a | log2FC| threshold of 1 and only genes with an adjusted p-value < 0.05 are shown. Select genes are labelled. $n = 3$ per group **c** Over-representation analysis of significant genes associated with each cell line. **d** Inferred activity of pathways predicted to be differentially active between the two models. Activity scores were calculated using PROGENy[27]. **e** Predicted Inferred activity of transcription factors predicted to be differentially active between the two models. Activity scores were calculated based on regulons from DoRothEA[48]. **f** Identical plot to (**b**), highlighting collagens (red), glycoproteins (blue), and proteoglycans (green) from the MatrisomeDB[49]. **g** Identical plot to (**b**), highlighting cytokines and their receptors from the KEGG pathway "Cytokine-cytokine receptor interaction". **h** Predicted proportion of tumour-associated macrophages (TAMs) among the non-malignant fraction of IB tumours. Proportions were based on bulk RNA-seq deconvolution using CIBERSORTx.

first PC reflected a differentiation axis that separated ascites-derived lines from the parental model (Fig. 5b, c). Ascites lines from the ID8 model (28-2, 30-2) lose activity in biosynthetic pathways (e.g., cholesterol, ribogenesis, and isoprenoid pathways), activating canonical EMT features, including signatures associated with ECM organisation, cytoskeleton remodelling, and TGFb signalling (Fig. 5d, Supplementary Fig. 2a). Notably, there was no particularly striking reduction in epithelial features in these ascites lines, consistent with a hybrid epithelial-mesenchymal phenotype[26, 36]. The activation of EMT features in STOSE ascites was less evident, perhaps due to the more-mesenchymal features of the parental lines, but a suppression of genes associated with epithelial morphogenesis was a prominent trend (Fig. 5e). Greenaway et al.[37] previously demonstrated that the ID8-derived ascites lines have an enhanced rate of migration compared to ID8 cells. Using a scratch wound assay, we demonstrated that the STOSE-A2 line also migrates more quickly than the parental STOSE line and produces abundant ascites in orthotopic tumour models (Supplementary Fig. 2b, c). Interestingly, STOSE-A3/4 did not migrate faster than STOSE cells, though we note that their position along the first principal component was less extreme than the A2 line (Fig. 5c), which may reflect a more epithelial phenotype. Despite resulting in more aggressive tumours, markers of proliferation were less abundant in ascites lines from both models (E2F Targets; Fig. 2d).

**Engineering mutations in tumour suppressor genes produce novel phenotypes that affect the TME.** Despite modelling various histological and molecular features of HGSOC, STOSE and ID8 models lack mutations commonly observed in the disease, such as Trp53, which is mutated in 96% of HGSOC cases[1]. To

better recapitulate the genetics of the disease, various clinically relevant mutations have been engineered into the ID8 model using the CRISPR-Cas9 system[20,21]. In our cohort of samples, we have included the ID8-Trp53−/− model along with four lines with a second inactivating mutation in either Brca1 (23% of HGSOC cases), Brca2 (11%), Nf1 (12%), or Pten (7%)[1] (Fig. 6a).

PCA of the cell line gene expression profiles highlighted the phenotypic divergence following the introduction of tumour suppressor mutations (Fig. 6b). Similar to the analysis of ascites-derived lines, the first PC seemed to reflect an axis of differentiation, with control, Trp53−/−, and Trp53−/−;Nf1−/− ID8 cells having evidence of elevated cholesterol biosynthesis and fructose 6-phosphate metabolism. The expression of these factors is suppressed in the Trp53−/−;Pten−/−, Trp53−/−;Brca1−/−, and Trp53−/−;Brca2−/− cells, which preferentially activate mesenchymal signatures (Fig. 6b). Among these less differentiated lines, we found that the Trp53−/−;Brca1−/− model expressed higher levels of monocyte chemotaxis factors whereas the Trp53−/−;Pten−/− model expressed inhibitors of chemotaxis, along with various ECM remodelling factors (collagen organisation, positive regulation of angiogenesis) (Fig. 6b).

We next evaluated whether these features may lead to distinct features in the TME of each model. Evaluating the bulk RNA-seq profile from whole tumour lysate for each model (Supplementary Fig. 3), we identified genes differentially expressed among the models and clustered genes based on their expression across models. Many clusters contained markers highly specific for non-malignant cell types, reflecting differences in the relative abundance of those cell types across models (Fig. 6c). Generic immune/defence signatures—likely associated with total immune infiltration—were notably deficient in the Pten−/− models (Fig. 6c). The Brca1−/− cell line expressed various monocyte

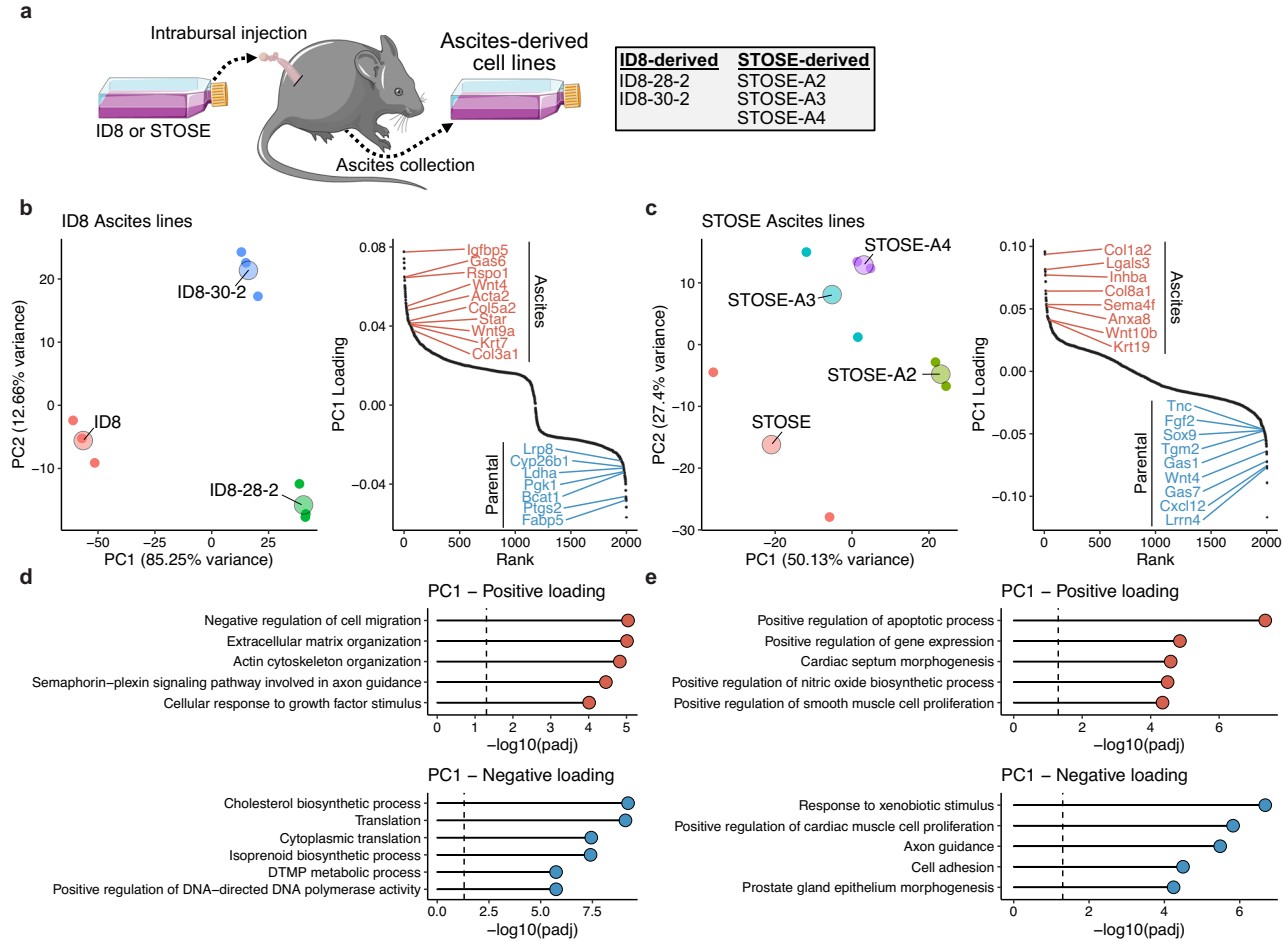

**Fig. 5 Activation of EMT-associated characteristics in ascites-derived lines. a** Schematic of ascites-derived subclones of ID8 and STOSE models. **b** PCA embedding (left) and ranked gene loadings for PC1 (right) for the ID8 models. **c** Same as (**b**) for STOSE-derived models. **d** Over-representation analysis of GO terms associated with the top and bottom 200 loaded genes for PC1 in the ID8 model. **e** Same as (**d**) but for STOSE-derived models.

recruitment factors in vitro and the $Brca1^{-/-}$ and $Brca2^{-/-}$ tumours had evidence of antigen processing expression in their bulk RNA profiles (Fig. 6c). Increased chemokine expression and macrophage infiltration in BRCA-mutant tumours is consistent with recent observations following the addition of a $Brca1^{-/-}$ mutation to an OVE-derived model engineered with $Trp53^{R172H}$, $Pten^{-/-}$, $Nf1^{-/-}$, and $Myc^{OE}$ mutations[15]. $Pten^{-/-}$ tumours, and to a lesser extent the $Nf1^{-/-}$ model, had increased expression of angiogenic signatures, further matching in vitro expression patterns (Fig. 6c). The absence of expression patterns suggestive of immune infiltration in $Pten^{-/-}$ tumours also supports the observation that human HGSOC tumours with intact $PTEN$ have an increased abundance of intraepithelial M2-like macrophages[38]. Finally, the $Nf1^{-/-}$ model had a unique enrichment of diverse immunoglobulins, reflecting an abundance of B cells (Fig. 6c).

## Discussion
Preclinical models are an invaluable resource to further our understanding of ovarian cancer, enabling the development of strategies to improve patient care. The number of ovarian cancer models is expanding and it has become increasingly unclear which models are most appropriate for specific experimental objectives. Here, we have focused on cataloguing and comparing transcriptomic data from a cohort of syngeneic mouse models of HGSOC.

Just as the molecular features of human tumours are diverse, these models displayed remarkable variation. While there has

been a great focus on deriving models with mutations that recapitulate the genetics of human HGSOC, the lack of recurrent mutations (beyond $Tp53$) and the diverse copy number alterations observed in human tumours make this goal intractable and perhaps irrelevant. This also raises the question of whether it's feasible to develop therapies that are selective to the genetics of the ~50% of tumours that are HR proficient. Effort in this direction is certainly critical and it is possible that a collection of targeted therapies could be utilised in conjunction with genetic screening to deliver personalised treatments to these patients. However, the challenges introduced by these complex genetics could be curtailed by focusing on developing therapeutics that are not dependent on them. For example, perturbations to signalling pathways may be used to sensitise chemoresistant phenotypes[39], or immunotherapies may be delivered to rejuvenate immune activity within the TME. Certainly, there has been an increased effort in this direction over the recent years. Therefore, there is an urgency to prioritise models based on their ability to recapitulate the general phenotypes and structural properties of human disease rather than the specific genetics.

OSE-derived models have received criticism for not reflecting the likely cell of origin for the majority of HGSOC. While this criticism is valid, it should not discount the information that these models have provided or the applicability of these models in future research. Both STOSE and ID8 are capable of producing ovarian tumours with histological features similar to human HGSOC[18,19], and so they—like all models—exhibit various

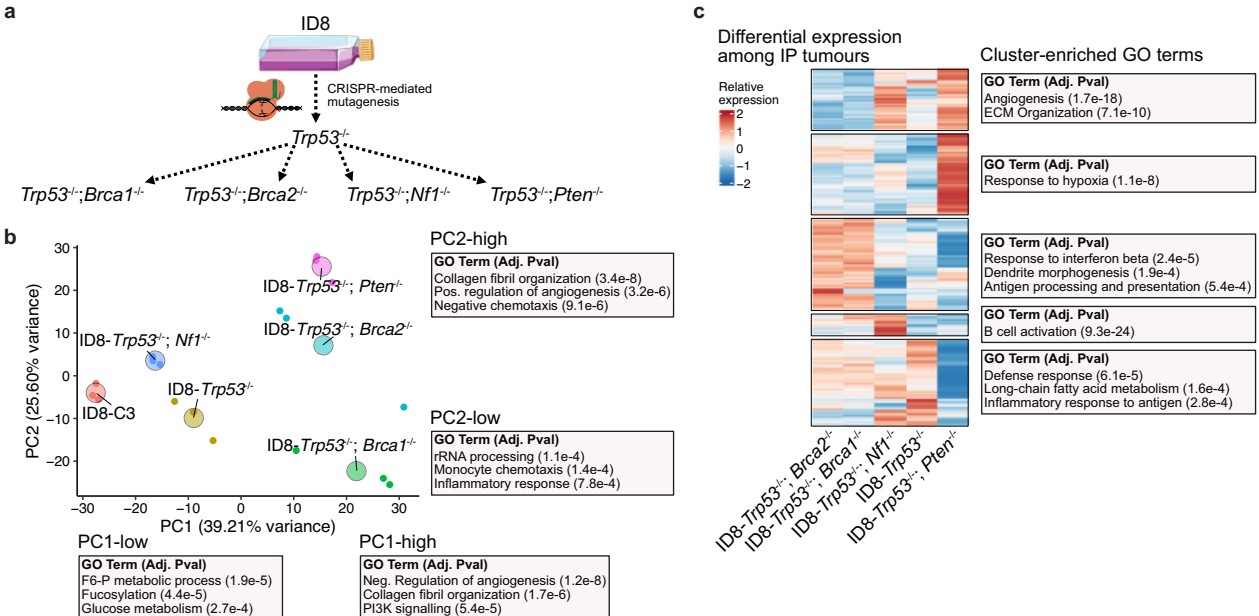

**Fig. 6 Variation among IP tumours of the ID8 model engineered with clinically relevant mutations. a** Schematic of the mutant ID8 lines. All mutations were previously engineered with CRISPR-mediated mutagenesis[20,21]. **b** PCA embedding of the transcription profile from engineered ID8 cell lines. Small points represent individual replicates of each model and the large labelled point is at the average coordinates for each model. Over-represented GO terms among top and bottom loaded genes for both PC1 and PC2 are shown next to the embedding along with Benjamini–Hochberg-adjusted *p*-values.
**c** Clustered heatmap of the 1544 genes differentially expressed (likelihood ratio test; adj. *p*-value < 0.05; *n* = 3–4 per group) among tumours from the ID8 models. Selected enriched GO terms for each cluster are shown next to each cluster.

features that are consistent with human disease and others that are not. Importantly, they produce tumours with distinct properties (e.g., the characteristically low stromal content of STOSE IB tumours), providing an opportunity to test therapeutics against diverse TMEs similar to those seen in human tumours. Similarly, although clinically relevant mutations have been engineered into the ID8 model, it is unclear if the phenotypic differences are strictly related to specific mutations that have been engineered or if they reflect divergence that occurred throughout the engineering process. It is perhaps more relevant that this has produced distinct sub-models with unique TME properties, such as the vascular rich *Pten*$^{-/-}$ model or the macrophage-rich *Brca1/2*$^{-/-}$ models.

Given the importance of recapitulating the TME of human disease, the experimental strategy for generating tumours is particularly relevant. We demonstrated that orthotopic tumours generated from intrabursal injection have more abundant tumour stroma than solid masses collected following intraperitoneal injection. This may simply be due to differences in tissue properties (e.g., stiffness, adiposity, etc.) between the ovary and sites within the peritoneal cavity (e.g., omentum). IP injection is experimentally convenient and often justified as a model of metastatic disease, but it is unclear if the resultant lesions faithfully recapitulate those emerging from the natural metastatic cascade from the ovary to these sites. Specifically, IP injection fails to impose the selective bottleneck that enriches for cells capable of leaving the adnexa. Rather, it gives the entire malignant population the opportunity to seed metastases, including cells that may not be capable of dissemination from a primary tumour. In our analysis, we demonstrated that ascites-derived cells are quite distinct from the parental population. It is unclear if this suggests that phenotypic selection is critical to metastatic dissemination or if the peritoneal cavity simply promotes phenotypic reprogramming.

While we present a relatively large cohort of syngeneic models, this list is by no means exhaustive and further work is required to

examine additional models. The addition of other data modalities (e.g., epigenetic profiling, deeper genetic characterisation, scRNA-seq of tumour models, etc.) would also provide valuable information about these models. Together, these initial comparisons have provided insight into phenotypic differences between the various models that ultimately affect the properties and progression of the tumours they make. Further validation of specific functional properties is certainly critical; however, we anticipate that this data will be particularly useful to aid in the selection of appropriate models for specific experimental aims.

## Methods

**Cell lines**. Unmodified ID8 cells were provided by Kathy Roby[19]. ID8-*Trp53*$^{-/-}$ F3, ID8-*Trp53*$^{-/-}$*Brca1*$^{-/-}$, ID8-*Trp53*$^{-/-}$*Brca2*$^{-/-}$, ID8-*Trp53*$^{-/-}$*Nf1*$^{-/-}$, ID8-*Trp53*$^{-/-}$*Pten*$^{-/-}$, were generated by CRISPR-Cas9 mediated knockout, and ID8-C3 (CRISPR control) were generously provided by Dr. Iain McNeish (Imperial College London)[20,21]. ID8-WT and its derivatives were maintained in Dulbecco's Modified Eagle Medium (DMEM, Corning) + 4% fetal bovine serum (FBS, Hyclone) and 0.01 mg/mL insulin-transferrin-sodium-selenite solution (ITSS; Roche) as previously described[20,21]. As previously described[37], ID8 ascites-derived lines (28-2, 30-2) were generated by culturing adherent cells from ascites formed in tumour-bearing mice approximately 60 days following orthotopic injection of parental ID8 cells. Cells were passaged 4–6 times to stabilise in vitro before being used for subsequent experiments.

STOSE cells were generated previously[18] and STOSE ascites-derived (STOSE-A2, A3, A4) cells were derived from ascites collected from three STOSE-tumour-bearing mice at the endpoint following intrabursal injection (approximately 74 days). STOSE-A cells were cultured for 20 passages to stabilise in vitro before being used for subsequent experiments. STOSE and STOSE-A cell lines were maintained in a-MEM (minimal essential media) + 4% FBS and ITSS and 2 μg/mL epithelial growth factor (EGF, R&D). OVE4 and OVE16 were oviductal epithelial cell lines generated

previously[17]. OVE4 were modified by Dr. Joanna Burdette (University of Illinois, Chicago) to generate MOE-$Pten^{shRNA}$, MOE-$Pten^{shRNA}$:$TP53^{R273H}$, and MOE-$Pten^{shRNA}KRAS^{G12V}$, and were maintained in a-MEM + 4% FBS, ITSS, 2 μg/mL EGF, and 18.2 ng/mL β-estradiol as previously described[22]. OVE4 and OVE16 with $Trp53^{-/-}$ or $TP53^{R175H}$ modifications were previously described[34]. Briefly, OVE4 and OVE16 cells were transfected with two pSp-Cas9 vectors each harbouring unique sgRNAs targeting the wild $Trp53$ locus. Effective knockout was confirmed by both Sanger sequencing and Western blot[34]. The $TP53^{R175H}$ modification was added to $Trp53^{-/-}$ cells by transducing lentivirus containing an expression vector for human $TP53^{R175H}$. Cells were incubated at 37 °C with 5% carbon dioxide.

To ensure the validity of cell lines, we employ several practices. First, the capacity to form tumours in immune-competent mice ensures that the cells are of the appropriate strain. Engineered modifications can be directly assessed by genotyping or Western blot. Finally, only low passage cells(split fewer than approximately 10 times following their receipt) are used for experiments, reducing the effects of phenotypic or genetic drift.

**Generation of tumours with syngeneic models**. Animal experiments to generate orthotopic tumours were carried out using protocols approved by the Animal Care Committee at the University of Ottawa and conforming to the standards defined by the Canadian Council on Animal Care (CCAC). FVB/N mice (for STOSE and MOE cell lines) were acquired from Charles River Laboratories and C57BL/6 mice (for ID8 and derivatives) were purchased from The Jackson Laboratory. Orthotopic intrabursal (IB) tumours were generated by injecting $1.5 \times 10^5$ cells (ID8-WT, ID8-C3,ID8-$p53^{-/-}$, STOSE, MOE-PTEN/p53, or MOE-PTEN/KRAS) under the ovarian bursa in 2 μL phosphate-buffered saline (PBS) as previously described[18]. Primary tumours were collected when mice reached humane endpoint, snap-frozen in liquid nitrogen, and stored at −80 °C until RNA extraction.

Procedures to generate IP tumours were approved by the University Animal Care Committee (UACC) at Queen's University and the CCAC. The various modified ID8 cell lines ($5–6 \times 10^6$, $n = 5–10$ per genotype) in 350 μL of PBS were transplanted via intraperitoneal injections into C57BL/6 female mice aged 8 to 10 weeks (Charles River Laboratories). Given the diffuse spread of the tumours, differences associated with the anatomic site were controlled for by selecting masses from the uterine horn for all models. Small miliary masses throughout the peritoneal cavity were not isolated. Adipose tissue was removed prior to dissociation.

All mice were maintained in specific pathogen-free conditions. Mice were left untreated until they reached their endpoint, which for IP tumours was deemed the point at which their abdominal diameters were approximately 34 mm. The tumours were snap-frozen in liquid nitrogen and stored at −80ºC until RNA extraction.

**In vivo carboplatin sensitivity**. Mice were injected intraperitoneally with $5 \times 10^6$ tumour cells resuspended in 100 μL PBS. Then, after 25% of the median survival times for each murine model (MOE-$Pten^{shRNA}$;$TP53^{R273H}$ treated at day 24; ID8-$Trp53^{-/-}$ at day 14; ID8-$Trp53^{-/-}$;$Brca2^{-/-}$ at day 14), mice received bi-weekly injections of carboplatin (Accord Healthcare Inc; provided by The Ottawa Hospital Pharmacy) at 20 mg/kg per mouse for a total of 4 weeks (8 doses). Control groups received saline. Mice were followed up for survival assessment until they reached humane endpoint.

**Western blot**. Cell suspensions were pelleted by centrifugation (5 min at $400 \times g$) after being washed two times with phosphate-buffered saline (PBS). They were then lysed with a master mix made of RIPA buffer (ThermoFisher, 89901), protease inhibitor (Sigma-Aldrich, P8340) and phosphatase inhibitor (Sigma-Aldrich, P0044) on ice for 15 min, then centrifuged at 15,000 rpm for 15 min at 4 °C. The supernatant was collected and quantified using Bio-Rad Protein Assay Dye Reagent concentrate diluted in water at a ratio of 1:5. A mixture of the concentrated protein, water and loading buffer with β-mercaptoethanol was prepared for each sample in order to have an equal amount of protein, and was then boiled for 5 min. Samples were separated on a NuPage 4–12% Bis-Tris gel (ThermoFisher, NP0336BOX) and then transferred into a PVDF membrane. After blocking using bovine serum albumin (BSA) for 1 h, the membrane was probed for ERK 2 (1:2000, SC-154, SantaCruz) and phosphor-ERK (1:2000, SC-7383, SantaCruz) antibodies overnight at 4 °C. The membrane was washed in TBST and then incubated for 1 h at room temperature in anti-mouse (1:10,000, ab6728, Abcam) and anti-rabbit (1:10,000, 711-035-152, Jackson Immunoresearch) secondary antibodies. The membrane was incubated for 1 h at room temperature in the loading control α-Tubulin (1:1000, 2144S, Cell Signaling). The relative abundance of protein was visualised using clarity western ECL (1705061, Bio-Rad).

**Quantitative reverse transcription polymerase chain reaction (RT-qPCR)**. Cells were released from adherent cultures using trypsin (0.05% trypsin, 0.53 mM EDTA) and washed with PBS before lysis in RLT Plus Lysis buffer (Qiagen) RNA was extracted according to manufacturer's instructions with the RNeasy Plus Mini Kit (Qiagen). RNA was then quantified, and cDNA was prepared with LunaScript RT SuperMix Kit (NEB), using 1 μg of RNA. Relative gene expression using primers (Invitrogen) specific to the genes of interest (Supplementary Table 1) was subsequently determined by qPCR using the SsoAdvanced Universal SYBR Green Supermix (Bio-Rad) on an Applied Biosystems 7500 Fast Real-Time PCR instrument. Gene expression was calculated as fold increase over untreated cells and normalised to the housekeeping genes $Ppia$ and $Rplp0$.

**Migration**. For migration assays, cells were plated at confluence (8000 STOSE and ascites-derived cells or 4000 ID8 cells) in each well of a 96-well plate (IncuCyte™ 96-Well ImageLock™ Plates, Essen Bioscience). Identical scratch wounds were made in every well using the IncuCyte WoundMaker (Essen BioScience, USA). Images were taken every 2 h with the IncuCyte automated monitoring system (Essen BioScience, USA), and IncuCyte ZOOM software was used to calculate wound confluence.

**RNA collection and library preparation**. Total RNA was extracted according to the manufacturer's instructions with the RNeasy Plus Mini Kit (Qiagen) for the majority of samples. IP tumours collected from mice injected with ID8-$Trp53^{-/-}$, ID8-$Trp53^{-/-}$; $Brca1^{-/-}$, ID8-$Trp53^{-/-}$; $Brca2^{-/-}$, ID8-$Trp53^{-/-}$; $Nf1^{-/-}$, and ID8-$Trp53^{-/-}$; $Pten^{-/-}$ cells were isolated at endpoint using the total RNA Purification Kit (Norgen Biotek Corporation, ON, Canada) as per the manufacturer's instructions.

RNA-Seq libraries were generated from 250 ng of total RNA as follows: mRNA enrichment was performed using the NEBNext Poly(A) Magnetic Isolation Module (New England BioLabs). cDNA synthesis was achieved with the NEBNext RNA First Strand Synthesis and NEBNext Ultra Directional RNA Second Strand Synthesis Modules (New England BioLabs). The remaining steps of library preparation were done using and the NEBNext Ultra II DNA Library Prep Kit for Illumina (New England

BioLabs). Adaptors and PCR primers were purchased from New England BioLabs.

The libraries were normalised and pooled and then denatured in 0.02 N NaOH and neutralised using HT1 buffer. The pool was loaded at 200pM on an Illumina NovaSeq S4 lane using Xp protocol as per the manufacturer's recommendations. The run was performed for $2 \times 100$ cycles (paired-end mode). A phiX library was used as a control and mixed with libraries at a 1% level. Base calling was performed with RTA v3. Program bcl2fastq2 v2.20 was then used to demultiplex samples and generate fastq reads.

**RNA-seq transcript quantification and processing**. Pseudo-lignment and transcript quantification for each sample were performed using Kallisto (v0.45.0)[40] with the GRCm38 build of the mouse genome. The R package tximport (v1.24.0) was used to load transcript quantifications, converting to gene-level transcript estimates. Principal component analysis (PCA) was performed on zero-centred normalised counts from DESeq2's variance stabilising vst() transformation. For each independent PCA, only the top 2000 variable genes were used as input. A t-distributed stochastic neighbour embedding (tSNE) was generated for the whole dataset using the top 10 principal components as input into the Rtsne() function provided by the Rtsne R package (v0.16).

**Differential gene expression**. All differential gene expression analysis was performed using DESeq2 (v1.36.0)[41]. To compare STOSE and ID8 cell lines, the Wald test was used to compute *p*-values and log fold change shrinkage was performed using the apeglm estimator[42]. To evaluate differentially expressed genes among more than two samples (e.g., the collection of modified ID8 lines), a likelihood ratio test was used. Genes with a *p*-value less than 0.05 and a standard deviation across all tested samples of greater than 0.5 (effectively providing an effect size threshold) were then clustered based on their relative expression levels across samples.

**GO Term over-representation analysis**. Over-representation analysis of GO terms among differentially expressed genes was performed using the topGO (v.2.48.0) wrapper topGOtable() provided in the pcaExplorer R package (v2.22.0)[43]. The elim method implemented in topGO was used to reduce redundancy in the list of enriched gene sets.

**Gene set scoring and inference of signalling and transcription factor activity**. The R package singscore (v1.16.0)[44] was used to compute gene set activity scores for individual samples. Scores reflect a rank-based statistic from genes comprising each set similar to the Wilcoxon rank sum test. Relative scores between samples were calculated by standardising scores across samples with a *Z*-score transformation.

Signalling activity for STOSE and ID8 models was calculated using the R package PROGENy (v1.18.0)[27] based on the package's pretrained regression models of gene activity associated with 14 different signalling pathways. The top 500 genes of each model were used for calculating scores.

Transcription factor activity was inferred using the database of transcription factor targets in the DoRothEA package (v1.8.0)[45], only including associations with a confidence level of "A" or "B". The viper method (v.1.30.0)[46] was used to compute activity scores for each individual sample. To compare activities between STOSE and ID8 samples, a general linear model was used for each factor. Only factors with a Benjamini–Hochberg-adjusted *p*-value of less than 0.05 were used.

**Bulk RNA-seq deconvolution using scRNA-seq**. Cell type deconvolution was performed using publicly available reference scRNA-seq data of primary orthotopic tumours from both STOSE and ID8 models[13] (NCBI GEO Accession: GSE183368). The annotated scRNA-seq data was first used to generate cell type references with CIBERSORTx[23], which was then used to deconvolve the normalised (TPM) quantifications for each bulk RNA-seq tumour sample using the default parameters. Predicted cell type proportions were then compared between samples.

**Reporting summary**. Further information on research design is available in the Nature Portfolio Reporting Summary linked to this article.

## Data availability

All RNA-seq data is available at the NCBI GEO Accession GSE242835. The source data behind the graphs in the figures can be found in Supplementary Data 1.

## Code availability

All code required to reproduce the analysis and figure production is available at https://github.com/dpcook/rna_seq_ovcan[47].

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

## Acknowledgements

Funding was provided by Health Canada to Ovarian Cancer Canada in support of the OvCAN research initiative. We would like to thank Genome Quebec for their assistance in generating the RNA-seq libraries. Several open access graphics (https://bioicons.com) have been used throughout the manuscript: mouse-darkgray (Figs. 1a, 4a, 5a), mouse-small (Fig. 4a), and culture-flask-filled-lid (Figs. 4a, 5a, 6a) icons are by Servier (https://smart.servier.com/) and are licensed under CC-BY 3.0 Unported https://creativecommons.org/licenses/by/3.0/; The organoid icon (Fig. 1a) by Marcel Tisch (https://twitter.com/MarcelTisch) is licensed under CC0 https://creativecommons.org/publicdomain/zero/1.0/; and the CRISPR-CAS9-pink icon (Fig. 6a) by DBCLS (https://togotv.dbcls.jp/en/pics.html) is licensed under CC-BY 4.0 https://creativecommons.org/licenses/by/4.0/.

## Author contributions

D.P.C., A.T., J.P., M.K., and B.C.V. conceived the study. K.G., G.M.R., N.S., J.W.S., M.P., M.E., K.M., J.H., and T.G.S. generated cell lines, performed all cell cultures, and collected RNA for library preparation. K.G., G.M.R., E.M., H.M., N.S., J.W.S., M.P., A.O.C., and K.M. performed animal studies. D.P.C. performed all computational analysis and drafted the manuscript. All authors revised and finalised the manuscript.

## Competing interests

The authors declare no competing interests.
