## [Peer Review File · Communications Biology]

Reviewers' comments:

Reviewer #1 (Remarks to the Author):

The ovarian cancer field is plagued with difficult interpretation and adaption of results stemming from mouse models. Such models are potentially valuable tools in understanding the complex interplay between tumours and their host. Unfortunately, however, ovarian cancer models are, as a group, poorly defined, if not poorly conceived (see below). Cook et al. attempt to rectify this deficiency by providing detailed transcriptomic analysis of several syngeneic "ovarian cancer models". These comprise lines derived from ovarian surface epithelium (ID8 and STOSE) and oviductal epithelium (OVE4 and derivatives) and genetically engineered variants of these lines. The transcriptomes are assessed for cells in culture, for tumors arising from ovarian bursa injections, and for tumors arising from intraperitoneal injections.

The work is, overall, technically sound, and the data may well prove useful to investigators who use these lines. For these reasons, I am not opposed to publication. However, I fundamentally believe that these lines are irrelevant to human ovarian cancer, and therefore should not be used to model the disease. As the authors note, ID8 and STOSE cells are derived from ovarian surface epithelium (OSE). It is now generally agreed that OSE is NOT the major cell of origin of human high grade serous ovarian carcinoma (HGSOC). Even worse, the mutations in the engineered variants are introduced into cells already tumorigenic by uncertain means. This in no way resembles what happens in the human disease, where the mutations drive tumorigenesis. Hence, ID8 and STOSE cells represent that uncommon/wrong cell-of-origin engineered post hoc with mutations that normally would drive tumor evolution (and the accompanying TME). Notably, the authors find that the cell-of-origin is a major determinant of the transcriptome (confirming the previous work of Zhang et al, Nat Comm 2019, which is not cited), which provides a further reason for concern about the relevance of OSE-derived syngeneic models.

The fallopian tube-derived models are potentially more relevant, yet again, these models are made by engineering already tumorigenic cells. Moreover, their characterization is far less extensive than that of the OSE models. Most importantly no effort is made to compare any of these models with the cognate human disease, for which ample transcriptional data is available.

In summary, I fully subscribe to the saying that "all models are wrong, but some are useful". Where one comes down on this paper, then, is determined by how useful you find these models. I think that they are more likely to misinform than inform, but I know that others (including the authors) hold quite the opposite position. I therefore support publication of this paper, but caveat emptor! In any event, attention should be given to the following issues before publication:

1) I think most readers—even those in the field—are not aware of the genesis of these models. I think their origin and engineering should be described in the first paragraph of Results.

2) The first line of the current first part of the Results section contains the following statement: "... we performed RNA-seq on a collection of commonly used mouse ovarian cancer cell lines.". Considering that only the ID8 and derivative models are, in fact, commonly used, and the others are models generated by the lab presenting this study, I find that statement a bit misleading. Instead, as they state in other parts of the manuscript, it would be better to state that these are a collection of mouse ovarian cancer cell lines.

3) The legend of Figure 1 contains the description of Figure 2E. I would encourage the authors to re-read the manuscript and try and find other easy-to-correct errors such as this one before final

submission.

4) The authors stipulate that they introduced Trp53R175H into OVE4 cells using a lentivirus. Although R175H is a major mutation in the human TP53 gene, the mouse Trp53 equivalent is R172H. Is this a simple error of notation of the authors or did they introduce the human gene into the mouse? Clarification is needed, especially as they do use the human TP53R273H mutant in some experiments.

5) Why did the authors choose to express mutant TP53 proteins in mice, while (presumably) keeping the WT protein intact? Wouldn't it had been more relevant to first KO WT Trp53 and then introduce mutant Trp53? If that is, in fact, what they did, they should make the Materials and Methods reflect this more clearly.

6) The authors describe using CRISPR-Cas9 to generate Trp53 KO. Before publication, the provenance of the pSpCas9-Trp53 plasmid should be noted, as well as the sgRNA sequences. The authors should make available the Western blots that validated the lines as part of supplementary information (both Trp53 KO and R172H OE), as well as the antibodies used and their conditions.

7) In the materials and methods, the authors don't mention the source of the carboplatin used in the in vivo injections, nor the carrier. This information should be provided. They should also explain what "25% of tumour establishment" means.

8) While the first figure indicates that 3 to 5 replicates of all RNA bulk experiments were performed, these numbers are not indicated again anywhere else. I suppose that similar numbers are used throughout the manuscript, but clarification is needed, in each figure and in the Materials and Methods. Indeed, all of the Figure legends are quite cursory.

9) Overall, this paper would benefit considerably from some simple in vitro experiments to validate some of the data presented, as as of now little is done in that direction. Some experiments that I would recommend performing:

a. At least a qPCR of selected genes (3 or more) from some ranked gene loadings, such as figure 2.B, 3.C, 4.B, 5.B, 5.C. Can be put in supplementary data.

b. Figure 4, a scratch assay would nicely validate 4.C, or a WB showing ERK1/2 activation. An ELISA would validate 4.G nicely too.

c. Similarly, in figure 5, a scratch assay or IF staining of actin (using phalloidin for example) with quantification of stress fibers would add a lot of strength to the data provided.

10) Figure 2, 5 and 6, the percentage of variation from component 1 and 2 should be indicated in the PCA plots.

11) Finally, this paper would largely benefit from some simple comparisons between the data provided here and relevant literature. Some of models here presented have been studied fairly extensively (as made clear by the references in the Introduction), so it should not be difficult for the authors to include a few sentences here and there comparing findings with that of other groups (or even their own). Such comparisons should be part of the discussion too, and would very much help put this research in the context of the broader field. Special interest should also be put in comparing the validity of these models with real human cases of HGSOE, as the end goal of these models is to mimic the human disease.

Reviewer #2 (Remarks to the Author):

In this manuscript, the authors present a comprehensive RNAseq analysis of cell lines and tumors from various murine models of ovarian high grade serous carcinoma. Their analyses include estimates of purity in the different models when injected ID and IP, differences between OSE and OVE models broadly, the role of Trp53 mutations, the divergence of the 'spontaneous' OSE models, the effect of utilising ascites-derived sublines and potential influence of specific clinically-relevant mutations in one model. In conjunction with their recent publication (Rodriguez et al, reference 13), this provides an extremely valuable dataset for ovarian cancer researchers and those utilizing these models (most of which are freely available).

Overall, the work has been performed to an extremely high standard. Although there is no overarching hypothesis, per se, this is high quality descriptive work that will influence multiple future studies and allow informed comparison and choice of models for e.g. therapeutic studies.

My comments are thus minor.

1. Use of CIBERSORT. Although CIBERSORT is an immensely useful tool, it was developed (and validated) using human datasets. Seq-ImmuCC is a murine specific tool (Chen et al DOI: 10.3389/fimmu.2018.01286) that might be of greater relevance here.
2. Figure 3 is very interesting. However, it might be possible to extend these data to correlate with human data further. The importance of the nature of TP53 mutations in high grade serous carcinoma is a controversial one. TCGA analysis (Hoadley et al 2014 – specifically Figure 5C) suggested that ovarian HGSC is marked loss of downstream WT p53 signalling regardless of mutation type (truncating vs missense). Do the various models assess here recapitulate that – are there differences between the R175H mutant and Trp53 null models in terms of specific p53 gene expression signatures both within cell lines and across the models?
3. Comparison with transgenic models. The data here all derive from transplantable models, the origins of which the authors describe here. Recently, transplantable models have been generated from Trp53^{-/-}; Brca2^{-/-}; Pten^{-/-} transgenic models (e.g. Maniati et al, Cell Rep 2020), with RNAseq data. It would be informative to include those if possible some form of comparative analysis if possible.

Reviewer #3 (Remarks to the Author):

The aim of this study is to perform comparative analysis of the transcriptome of different mouse models of high-grade serous ovarian cancer. Given the heterogeneity of the disease and numerous models available, the data obtained will be helpful in identification of relevant drivers based on cellular origin, anatomical location and mutational burden. The study is purely bioinformatics based and lacks validation at either protein or cellular level. Nonetheless, it is appreciated that the authors referred to previous studies that can support/validate the transcriptomic differences identified. Since validation is important, the following points need to be addressed. These are mostly clarifications on the methodology which are important for future reproducibility and validation.

1) For Fig. 5. Please expand the methods section and clarify how many days after injection ascites was collected. Were the cells passed several times in culture before RNA seq? Include graphs of abdominal width and final tumor weight at endpoint. Comment on whether there were other sites of implantation other than IB.

2) Similarly for Fig. 6, please include graphs of abdominal width and final tumor weight at endpoint. Ip

tumors from ID8 cells are military and can be challenging to separate from normal tissues. Comment on how this was controlled for.

Minor:

- 1) Please clarify in Figure legends the groups that are being compared. Fig. 3 for instance, is each cell line compared to its parent?
- 2) Figure legends for Figure 1 includes Fig. 1E but the actual figure is missing and also not referred to in the text.
- 3) Standard statement on cell line integrity (such as STR tests) is missing.

Reviewer #1

The ovarian cancer field is plagued with difficult interpretation and adaption of results stemming from mouse models. Such models are potentially valuable tools in understanding the complex interplay between tumours and their host. Unfortunately, however, ovarian cancer models are, as a group, poorly defined, if not poorly conceived (see below). Cook et al. attempt to rectify this deficiency by providing detailed transcriptomic analysis of several syngeneic “ovarian cancer models”. These comprise lines derived from ovarian surface epithelium (ID8 and STOSE) and oviductal epithelium (OVE4 and derivatives) and genetically engineered variants of these lines. The transcriptomes are assessed for cells in culture, for tumors arising from ovarian bursa injections, and for tumors arising from intraperitoneal injections.

The work is, overall, technically sound, and the data may well prove useful to investigators who use these lines. For these reasons, I am not opposed to publication. However, I fundamentally believe that these lines are irrelevant to human ovarian cancer, and therefore should not be used to model the disease. As the authors note, ID8 and STOSE cells are derived from ovarian surface epithelium (OSE). It is now generally agreed that OSE is NOT the major cell of origin of human high grade serous ovarian carcinoma (HGSOC). Even worse, the mutations in the engineered variants are introduced into cells already tumorigenic by uncertain means. This in no way resembles what happens in the human disease, where the mutations drive tumorigenesis. Hence, ID8 and STOSE cells represent that uncommon/wrong cell-of-origin engineered post hoc with mutations that normally would drive tumor evolution (and the accompanying TME). Notably, the authors find that the cell-of-origin is a major determinant of the transcriptome (confirming the previous work of Zhang et al, Nat Comm 2019, which is not cited—**now cited, thank you for noting this**), which provides a further reason for concern about the relevance of OSE-derived syngeneic models.

The fallopian tube-derived models are potentially more relevant, yet again, these models are made by engineering already tumorigenic cells (**to clarify, parental OVE4 and OVE16 models are not tumorigenic—only the engineered derivatives are. We hope this is clear in the modified first paragraph of the results**). Moreover, their characterization is far less extensive than that of the OSE models. Most importantly no effort is made to compare any of these models with the cognate human disease, for which ample transcriptional data is available.

In summary, I fully subscribe to the saying that “all models are wrong, but some are useful”. Where one comes down on this paper, then, is determined by how useful you find these models. I think that they are more likely to misinform than inform, but I know that others (including the authors) hold quite the opposite position. I therefore support publication of this paper, but caveat emptor! In any event, attention should be given to the following issues before publication:

We would like to sincerely thank Reviewer 1 for their critical appraisal of this work. It is clear that the manuscript was read carefully, which is much appreciated. We recognize the limitations of models derived from the ovarian surface epithelium and fully support the adoption of newer models from the oviductal epithelium (eg. Iyer et al., 2021, which was published after we

initiated our study). However, we do believe data from OSE-derived models can still have merit, in part because current evidence suggests that at least some HGSOE derive from OSE, and because comparisons with fallopian tube-derived models can be particularly informative.

Comment 1-1

I think most readers—even those in the field—are not aware of the genesis of these models. I think their origin and engineering should be described in the first paragraph of Results.

Response 1-1

We have now provided additional detail about the models evaluated at the beginning of the results. We have added references to the source studies (were initially included only in methods) and have modified the text to provide additional information:

Lines 93-102

This collection comprises models from both oviductal (OVE/MOE¹⁷) and ovarian surface epithelium (OSE), spontaneously transformed (STOSE¹⁸ and ID8¹⁹) models, secondary lines derived from ascites, and derivative lines engineered with clinically relevant mutations in tumour suppressor genes or constitutive activation of oncogenes^{20–22}. For select models, we also sequenced RNA from tumours derived from either intrabursal (IB) or intraperitoneal (IP) injection of the cells, allowing us to evaluate how properties of the models and TME may affect features of the resultant tumours (**Figure 1A**). All evaluated models were initially derived from primary cultures of non-tumorigenic ovarian or oviductal epithelium. Tumorigenicity was acquired spontaneously through long-term propagation (STOSE and ID8) or repression of the tumor suppressors *Trp53* (OVE4 and OVE16 derivatives) or *Pten* (MOE).

Comment 1-2

The first line of the current first part of the Results section contains the following statement: "... we performed RNA-seq on a collection of commonly used mouse ovarian cancer cell lines.". Considering that only the ID8 and derivative models are, in fact, commonly used, and the others are models generated by the lab presenting this study, I find that statement a bit misleading. Instead, as they state in other parts of the manuscript, it would be better to state that these are a collection of mouse ovarian cancer cell lines.

Response 1-2

The ID8 models are certainly most common among the collection, but STOSE and OVE/MOE lines have been disseminated widely. We have modified the text to avoid claims of popularity.

Lines 91-92

To develop a resource of transcriptomic data that could provide further insight into models of HGSOE, we performed RNA-seq on a collection of mouse ovarian cancer cell lines (**Figure 1A**).

Comment 1-3

The legend of Figure 1 contains the description of Figure 2E. I would encourage the authors to re-read the manuscript and try and find other easy-to-correct errors such as this one before final submission.

Response 1-3

Apologies that we missed that during our final edits. We have gone through the manuscript to ensure figure references and legends match the appropriate panel.

Comment 1-4

The authors stipulate that they introduced Trp53R175H into OVE4 cells using a lentivirus. Although R175H is a major mutation in the human TP53 gene, the mouse Trp53 equivalent is R172H. Is this a simple error of notation of the authors or did they introduce the human gene into the mouse? Clarification is needed, especially as they do use the human TP53R273H mutant in some experiments.

Response 1-4

To confirm, human *TP53*^{R175H} was expressed in the OVE cells lacking wild-type Trp53 (KO'd via CRISPR). The MOE-PTEN-TP53^{273H} cells, however, were from Eddie et al., *Oncotarget*, 2015 and still harbor wild-type *Trp53* alleles. Apologies for the confusion and we have modified the methods accordingly and have modified the nomenclature in the text/figures to more clearly differentiate between *Trp53*^{-/-} and *TP53*^{R175H} modifications:

Lines 395-400

OVE4 and OVE16 with *Trp53*^{-/-} or *TP53*^{R175H} modifications were previously described³⁴. Briefly, OVE4 and OVE16 cells were transfected with two pSp-Cas9 vectors each harboring unique sgRNAs targeting the wild *Trp53* locus. Effective knock-out was confirmed by both Sanger sequencing and Western blot³⁴. The *TP53*^{R175H} modification was added to *Trp53*^{-/-} cells by transducing lentivirus containing an expression vector for human *TP53*^{R175H}. Cells were incubated at 37°C with 5% carbon dioxide.

Comment 1-5

Why did the authors choose to express mutant TP53 proteins in mice, while (presumably) keeping the WT protein intact? Wouldn't it had been more relevant to first KO WT Trp53 and then introduce mutant Trp53? If that is, in fact, what they did, they should make the Materials and Methods reflect this more clearly.

Response 1-5

Addressed in Response 1-4

Comment 1-6

The authors describe using CRISPR-Cas9 to generate Trp53 KO. Before publication, the provenance of the pSpCas9-Trp53 plasmid should be noted, as well as the sgRNA sequences. The authors should make available the Western blots that validated the lines as part of supplementary information (both Trp53 KO and R172H OE), as well as the antibodies used and their conditions.

Response 1-6

Since initial submission, another study involving these cells has been published (Haagsma *et al.*, *Sci Rep*, 2023) that includes validation of Trp53 KO (Western blot, Sanger sequencing) and R175H expression. We have cited this information in the methods (Response 1-4):

Comment 1-7

In the materials and methods, the authors don't mention the source of the carboplatin used in the in vivo injections, nor the carrier. This information should be provided. They should also explain what "25% of tumour establishment" means.

Response 1-7

Apologies—we could have been clearer. Given differences in growth rate and median survival for each model, animals were treated with carboplatin at 25% of the median survival time for each specific model, as determined from multiple previous studies, including Rodriguez *et al.*. Specific timing along with the source of our carboplatin have been included in the methods:

Lines 433-438

Mice were injected intraperitoneally with 5×10^6 tumor cells resuspended in 100 μ l PBS. Then, after 25% of the median survival time for each murine model (MOE-*Pten*^{shRNA}; *TP53*^{R273H} treated at day 24; ID8-*Trp53*^{-/-} at day 14; ID8-*Trp53*^{-/-}; *Brca2*^{-/-} at day 14), mice received bi-weekly injections of carboplatin (Accord Healthcare Inc; provided by The Ottawa Hospital Pharmacy) at 20mg/kg per mouse for a total of 4 weeks (8 doses). Control groups received saline. Mice were followed-up for survival assessment until they reached humane endpoint.

Comment 1-8

While the first figure indicates that 3 to 5 replicates of all RNA bulk experiments were performed, these numbers are not indicated again anywhere else. I suppose that similar numbers are used throughout the manuscript, but clarification is needed, in each figure and in the Materials and Methods. Indeed, all of the Figure legends are quite re.

Response 1-8

We have now included replicate numbers in the figure legends for all panels involving either statistical testing or where replicates are not explicitly shown (eg. because replicate-averaged values were shown instead). We have modified legends for Figure 1B, 2A/E, 3A, 4B, 6C to include this information.

Comment 1-9

Overall, this paper would benefit considerably from some simple in vitro experiments to validate some of the data presented, as as of now little is done in that direction. Some experiments that I would recommend performing:

a. At least a qPCR of selected genes (3 or more) from some ranked gene loadings, such as figure 2.B, 3.C, 4.B, 5.B, 5.C. Can be put in supplementary data.

Response 1-9a

We have now included qPCR validation of several expression changes throughout the manuscript on independent replicates from those used for RNA-seq.

Supplemental Figure 1A - Differences between STOSE and ID8 cells

Supplemental Figure 2A - Expression changes in ascites-derived lines

b. Figure 4, a scratch assay would nicely validate 4.C, or a WB showing ERK1/2 activation. An ELISA would validate 4.G nicely too.

Response 1-9b

We have now included a scratch assay comparing migration rates of STOSE and ID8 cells.

Supplemental Figure 1C

We have also included validation of increased pERK in STOSE, consistent with expression changes in Fig 3C.

Supplemental Figure 1B

c. Similarly, in figure 5, a scratch assay or IF staining of actin (using phalloidin for example) with quantification of stress fibers would add a lot of strength to the data provided.

Response 1-9c

We have now included a scratch assay of the STOSE ascites lines demonstrating enhanced migration in the STOSE-A2 line relative to the other models. This line also has the highest PC1 value in Fig 5C, consistent with the interpretation of the gene expression. We would have

predicted that the A3/A4 would have had an intermediate increase in migration, but this doesn't seem to be the case. We have added a brief discussion of this in the results

Lines 262-268

Greenaway et al.³⁷ previously demonstrated that the ID8-derived ascites lines have an enhanced rate of migration compared to ID8 cells. Using a scratch wound assay, we demonstrated that the STOSE-A2 line also migrates more quickly than the parental STOSE line and produces abundant ascites in orthotopic tumour models (**Supplemental Figure 2B,C**). Interestingly, STOSE-A3/4 did not migrate faster than STOSE cells, though we note that their position along the first principal component was less extreme than the A2 line (**Figure 5C**), which may reflect a more epithelial phenotype.

Supplemental Figure 2B- STOSE ascites migration

Comment 1-10

Figure 2, 5 and 6, the percentage of variation from component 1 and 2 should be indicated in the PCA plots.

Response 1-10

This information has now been added to all PCA plots.

Figure 2A

Figure 5B/C

Figure 6B

Comment 1-11

Finally, this paper would largely benefit from some simple comparisons between the data provided here and relevant literature. Some of models here presented have been studied fairly extensively (as made clear by the references in the Introduction), so it should not be difficult for the authors to include a few sentences here and there comparing findings with that of other groups (or even their own). Such comparisons should be part of the discussion too, and would very much help put this research in the context of the broader field. Special interest should also be put in comparing the validity of these models with real human cases of HGSOV, as the end goal of these models is to mimic the human disease.

Response 1-11

Throughout each major section of the manuscript, we have now tried to provide additional connections to existing literature.

Lines 123-127

This distinction based on cell-of-origin has also been previously observed in both engineered organoid models from OSE and OVE tissue²⁴ and genetically engineered mouse models of HGSOC²⁵, suggesting that although models from both tissues can form tumours with HGSOC characteristics, phenotypic differences of the malignant compartment exist.

Lines 149-150

These findings are also in line with previous work demonstrating that the ID8-*Trp53*^{-/-}; *Brca2*^{-/-} model is more sensitive to chemotherapy than ID8-*Trp53*^{-/-} tumours²¹.

Lines 167-171

Hoadley *et al.*³³ previously examined p53 signalling activity in human tumours harboring *TP53* missense mutations and, using the PARADIGM method to infer signalling activity, found that the majority of missense mutations in HGSOC result in decreased activity relative to several wild-type tumours. In contrast, we observed that expression of *TP53*^{R175H} in *Trp53*-null OVE4/16 cells restored the inferred p53 activity (**Figure 3A**).

Lines 176-178

OVE4 cells reactivated expression of various genes involved in damage responses (**Figure 3E**), which may relate to the previously observed relationship between the R175H mutation and apoptosis signalling in these cells³⁴.

Lines 183-188

While many of the specific genes varied, failure to re-activate several features of epithelial differentiation were common, such as the maintained suppression of various cell adhesion genes (eg. *Cldn1*, *Cldn4*, *Cdh1*) or genes associated with EGF signalling (**Figure 3E,F**). This putative dedifferentiation may relate to the various phenotypic properties of cells with *TP53*^{R175H} mutations, including enhanced metastatic capacity, stemness, and drug resistance³⁵.

Lines 240-242

These inferences about the TME of STOSE tumours and their more immunoregulatory phenotype compared to ID8 cells also match recent flow cytometry-based immunophenotyping and cytokine profiling of these models¹³.

Lines 262-263

Greenaway *et al.*³⁷ previously demonstrated that the ID8-derived ascites lines have an enhanced rate of migration compared to ID8 cells.

Lines 302-305

Increased chemokine expression and macrophage infiltration in BRCA-mutant tumours is consistent with recent observations following the addition of a *Brca1*^{-/-} mutation to an OVE-derived model engineered with *Trp53*^{R172H}, *Pten*^{-/-}, *Nf1*^{-/-}, and *Myc*^{OE} mutations¹⁵.

Lines 306-309

The absence of expression patterns suggestive of immune infiltration in *Pten*^{-/-} tumours also supports the observation that human HGSOC tumours with intact *PTEN* have an increased abundance of intraepithelial M2-like macrophages³⁸

Reviewer #2

In this manuscript, the authors present a comprehensive RNAseq analysis of cell lines and tumors from various murine models of ovarian high grade serous carcinoma. Their analyses include estimates of purity in the different models when injected ID and IP, differences between OSE and OVE models broadly, the role of Trp53 mutations, the divergence of the 'spontaneous' OSE models, the effect of utilising ascites-derived sublines and potential influence of specific clinically-relevant mutations in one model. In conjunction with their recent publication (Rodriguez et al, reference 13), this provides an extremely valuable dataset for ovarian cancer researchers and those utilizing these models (most of which are freely available).

Overall, the work has been performed to an extremely high standard. Although there is no over-arching hypothesis, per se, this is high quality descriptive work that will influence multiple future studies and allow informed comparison and choice of models for e.g. therapeutic studies.

My comments are thus minor.

Comment 2-1

Use of CIBERSORT. Although CIBERSORT is an immensely useful tool, it was developed (and validated) using human datasets. Seq-ImmuCC is a murine specific tool (Chen et al DOI: 10.3389/fimmu.2018.01286) that might be of greater relevance here.

Response 2-1

Although the original CIBERSORT method was demonstrated and validated using data from human tumours, there is nothing inherently species specific about the approach (ie. using a regression model to deconvolve a mixture using reference profiles of pure components). To our understanding, the concern of species specificity in the original CIBERSORT method was in the use of the LM22 reference data of human leukocytes provided by the authors. However, in our manuscript, we have used the updated version of the method (CIBERSORTx) that has been tuned to use scRNA-seq data as the reference for deconvolution. For this, we have specifically used scRNA-seq data from STOSE and ID8 tumours (Rodriguez et al., *Cancer Research Communications*, 2022). We believe this is the most sound approach for deconvolution given its direct relevance to the samples explored here.

Comment 2-2

Figure 3 is very interesting. However, it might be possible to extend these data to correlate with human data further. The importance of the nature of TP53 mutations in high grade serous carcinoma is a controversial one. TCGA analysis (Hoadley et al 2014 – specifically Figure 5C) suggested that ovarian HGSC is marked loss of downstream WT p53 signalling regardless of mutation type (truncating vs missense). Do the various models assessed here recapitulate that – are there differences between the R175H mutant and Trp53 null models in terms of specific p53 gene expression signatures both within cell lines and across the models?

Response 2-2

We have modified Figure 3 to more thoroughly compare the effects of null vs. missense mutations. We have now included a plot showing inferred p53 signalling activity in the models in Figure 3. As expected, we observe a relative decrease in activity in the null mutants. In contrast to the Hoadley finding, however, inferred activity tends to be restored in the R175H mutants. In general, we find that R175H expression restores a subset of effects associated with null mutations in both OVE4 and 16 models. Novel expression patterns associated with R175H expression are not overly abundant. We do, however, find that these effects are more prominent in OVE16, but the reason for this is unclear.

Additional note: running the PARADIGM method in Hoadley et al is no longer feasible as the source code license seems to have been transferred/sold to Five3 Genomics (paradigm.five3genomics.com) and cannot be accessed at the time of writing this response.

Figure 3

Comment 2-3

Comparison with transgenic models. The data here all derive from transplantable models, the origins of which the authors describe here. Recently, transplantable models have been generated from *Trp53*^{-/-}; *Brca2*^{-/-}; *Pten*^{-/-} transgenic models (e.g. Maniati et al, Cell Rep 2020), with RNAseq data. It would be informative to include those if possible some form of comparative analysis if possible.

Response 2-3

A valid integration of the RNA-seq data from Maniati et al would be particularly difficult given the inherent technical variation between the two studies. A common pattern described in their study is the difference between their HGS1-4 (FTE) and 30200/60577 model, consistent with our observation that cell-of-origin is the dominant source of variation among the models. We have included a brief discussion of this and reference to their study.

Lines 123-127

This distinction based on cell-of-origin has also been previously observed in both engineered organoid models from OSE and OVE tissue²⁴ and genetically engineered mouse models of HGSOC²⁵, suggesting that although models from both tissues can form tumours with HGSOC characteristics, phenotypic differences of the malignant compartment exist.

Additional comparisons are difficult to make as many features in their study are fairly consistent between the HGS1-4 models, making it difficult to assess impact of, for example, the impact of individual mutations.

Reviewer #3:

The aim of this study is to perform comparative analysis of the transcriptome of different mouse models of high-grade serous ovarian cancer. Given the heterogeneity of the disease and numerous models available, the data obtained will be helpful in identification of relevant drivers based on cellular origin, anatomical location and mutational burden. The study is purely bioinformatics based and lacks validation at either protein or cellular level. Nonetheless, it is appreciated that the authors referred to previous studies that can support/validate the transcriptomic differences identified. Since validation is important, the following points need to be addressed. These are mostly clarifications on the methodology which are important for future reproducibility and validation.

We appreciate the reviewer's comments and provide specific responses below. We would also like to note that we have added data in several places to provide validation for some of the inferences made from expression data. See Response 1-9 for further detail.

Comment 3-1

For Fig. 5. Please expand the methods section and clarify how many days after injection ascites was collected. Were the cells passed several times in culture before RNA seq? Include graphs of abdominal width and final tumor weight at endpoint. Comment on whether there were other sites of implantation other than IB.

Response 3-1

We have now included a brief statement specifying that ascites were collected at endpoint (approximately 60 days for ID8 and 74 days for STOSE). All of the ascites lines were derived from IB tumours.

Lines 381-388 (Methods)

As previously described³⁷, ID8 ascites-derived lines (28-2, 30-2) were generated by culturing adherent cells from ascites formed in tumour-bearing mice approximately 60 days following orthotopic injection of parental ID8 cells. Cells were passaged 4-6 times to stabilize in vitro before being used for subsequent experiments. STOSE cells were generated previously¹⁸ and STOSE ascites-derived (STOSE-A2, A3, A4) cells were derived from ascites collected from three STOSE-tumour bearing mice at endpoint following intrabursal injection (approximately 74 days).

The ID8-derived lines were previously described in Greenaway et al., 2016. Abdominal width was not measured, but we have included a graph of tumour burden and ascites volume for the STOSE-derived ascites models.

Supplemental Figure 2C

Comment 3-2

Similarly for Fig. 6, please include graphs of abdominal width and final tumor weight at endpoint. Ip tumors from ID8 cells are miliary and can be challenging to separate from normal tissues. Comment on how this was controlled for.

Response 3-2

Apologies that sufficient detail was not included. We have provided additional information in the methods. The tumours collected from the IP injections were specifically masses from the uterine horn to help eliminate variation between samples. The small miliary tumours throughout the peritoneal cavity were avoided. We find the masses from the uterine horn are easier to separate from healthy tissue because of their density. Consistency in the replicate data also provide some confidence that technical variation is relatively minimal.

Lines 419-425 (Methods)

Procedures to generate IP tumours were approved by the University Animal Care Committee (UACC) at Queen's University and the CCAC. The various modified ID8 cell lines ($5-6 \times 10^6$, $n=5-10$ per genotype) in $350 \mu\text{L}$ of PBS were transplanted via intraperitoneal injections into C57BL/6 female mice aged 8 to 10 weeks (Charles River Laboratories). Given the diffuse spread of the tumours, differences associated with anatomic site were controlled for by selecting masses from the uterine horn for all models. Small miliary masses throughout the peritoneal cavity were not isolated. Adipose tissue was removed prior to dissociation.

We have also included measurements of tumour burden and ascites volume for the various ID8 IP models.

Supplemental Figure 3A

Minor:

Comment 3-3

Please clarify in Figure legends the groups that are being compared. Fig. 3 for instance, is each cell line compared to its parent?

Response 3-3

While Figure 3 has been modified, we have ensured that comparisons for differential expression is clear in the figure legends.

Figure 3 Legend

Figure 3. Common and divergent effects of null and missense p53 mutations. A. Inferred p53 signalling activity in lines with wild type, nonsense, and missense p53. Activity values are Z-scores derived by PROGENY(Schubert et al. 2018). **B.** Pairwise intersection of differentially expressed genes ($p < 0.05$, $|\log_2FC| > 1$; $n=3$) associated with *Trp53*^{-/-} mutations relative to parental lines. **C.** Frequency of genes upregulated or downregulated upon *Trp53* deletion compared to parental lines. **D.** Expression values of select genes commonly upregulated (*Map1a*, *Prr5l*, *Rcn3*) or downregulated (*Ngf*, *Cdkn1a*, *Areg*) following *Trp53* deletion. Expression values reflect log-transformed transcripts per million (logTPM). **E.** Comparison of expression patterns in OVE4 cells following *Trp53* deletion (p53 KO vs. wild-type) or re-expression of the missense *TP53*^{R175H} (R175H vs. KO). Select genes and GO terms (p-values included in parentheses) are included. **F.** Same as (E) for OVE16 cells.

Comment 3-4

Figure legends for Figure 1 includes Fig. 1E but the actual figure is missing and also not referred to in the text.

Response 3-4

Apologies—that caption belonged to the survival curves in Figure 2E. Legends have been checked and modified accordingly.

Comment 3-5

Standard statement on cell line integrity (such as STR tests) is missing.

Response 3-5

We now include a statement on cell line integrity in the methods. Given that all the models come from one of two inbred mouse strains, STR tests are not particularly informative. We try to ensure validity of the lines through several approaches: 1) the capacity to form tumours is dependent on the strain of the cells matching the recipient mouse, and we can therefore confirm that models are consistently the appropriate background; 2) to avoid issues with phenotypic/genetic drift, we maintain stocks of cells from early passages immediately after they are received and only cells from low passage (<10) are used for experiments; 3) engineered modifications (eg. gene deletion) can be directly verified via genotyping or Western blot.

Lines 402-406 (Methods)

To ensure the validity of cell lines, we employ several practices. First, the capacity to form tumours in immune-competent mice ensures that the cells are of the appropriate strain. Engineered modifications can be directly assessed by genotyping or Western blot. Finally, only low passage cells (split fewer than approximately 10 times following their receipt) are used for experiments, reducing the effects of phenotypic or genetic drift.

REVIEWERS' COMMENTS:

Reviewer #1 (Remarks to the Author):

I would like to congratulate the authors of this paper for successfully resolving the potential issues I pointed in their original submission. I am happy to report that they have gone above and beyond my expectations while tackling them, and in doing so have created a much stronger publication which I believe they can be very proud of. This paper will surely be a trove of very valuable information for researchers using these models, and surely will help advance the field of HGSOC research in the right direction!

I recommend this manuscript be published as is without any further changes needed.

Reviewer #2 (Remarks to the Author):

I raised three relatively minor comments in my original review, relating to the use of CIBERSORT, comparisons of p53 null vs p53 missense and transgenic models.

The authors have answered these clearly and succinctly.

I have no further concerns.